# Effects of Combining a Ketogenic Diet with Resistance Training on Body Composition, Strength, and Mechanical Power in Trained Individuals: A Narrative Review

**DOI:** 10.3390/nu13093083

**Published:** 2021-09-01

**Authors:** Pedro L. Valenzuela, Adrián Castillo-García, Alejandro Lucia, Fernando Naclerio

**Affiliations:** 1Faculty of Sport Sciences, Universidad Europea de Madrid, 28670 Madrid, Spain; pedroluis.valenzuela@universidadeuropea.es (P.L.V.); alejandro.lucia@universidadeuropea.es (A.L.); 2Fissac—Physiology, Health and Physical Activity, 08015 Barcelona, Spain; adriancastillogarcia@icloud.com; 3Physical Activity and Health Research Group (‘PaHerg’), Research Institute of the Hospital 12 de Octubre (‘imas12’), 28041 Madrid, Spain; 4Institute for Lifecourse Development, School of Human Sciences, Centre for Exercise Activity and Rehabilitation, University of Greenwich, London SE10 9LS, UK

**Keywords:** low-carbohydrate, power output, resistance training, muscle, keto

## Abstract

Ketogenic diets (KD) have gained popularity in recent years among strength-trained individuals. The present review summarizes current evidence—with a particular focus on randomized controlled trials—on the effects of KD on body composition and muscle performance (strength and power output) in strength-trained individuals. Although long-term studies (>12 weeks) are lacking, growing evidence supports the effectiveness of an ad libitum and energy-balanced KD for reducing total body and fat mass, at least in the short term. However, no or negligible benefits on body composition have been observed when comparing hypocaloric KD with conventional diets resulting in the same energy deficit. Moreover, some studies suggest that KD might impair resistance training-induced muscle hypertrophy, sometimes with concomitant decrements in muscle performance, at least when expressed in absolute units and not relative to total body mass (e.g., one-repetition maximum). KD might therefore be a beneficial strategy for promoting fat loss, although it might not be a recommendable option to gain muscle mass and strength/power. More research is needed on the adoption of strategies for avoiding the potentially detrimental effect of KD on muscle mass and strength/power (e.g., increasing protein intake, reintroduction of carbohydrates before competition). In summary, evidence is as yet scarce to support a major beneficial effect of KD on body composition or performance in strength-trained individuals. Furthermore, the long-term effectiveness and safety of this type of diet remains to be determined.

## 1. Introduction

Ketogenic diets (KDs) aim at inducing physiological ketosis (i.e., an increase in the concentration of ketone bodies in blood, usually above >0.5 mmol/L) through a marked reduction in carbohydrate intake (commonly <50 g/d or <10% of total energy intake) [1]. KDs have gained popularity in recent years among athletes [2]. By virtue of the restriction they induce in carbohydrate availability, KDs promote the use of ketone bodies (i.e., acetoacetate, acetone and β-hydroxybutyrate (BHB)) as an alternative energy substrate for different body tissues. Ketone bodies are produced from free fatty acids mainly in the mitochondria of liver cells. Once in the bloodstream, acetoacetate and BHB (the two ketone bodies used for energy) can reach extrahepatic tissues (notably, skeletal muscles, heart, brain). BHB is converted to acetoacetate by a reaction catalyzed by BHB dehydrogenase, and acetoacetate is converted back to acetyl-CoA by the action of a beta-ketoacyl-CoA transferase. The resulting acetyl CoA enters the Krebs cycle to produce ATP through the electron transport chain. Due to the initial, non-energy demanding activation of ketone bodies into an oxidable form (in a reaction catalyzed by succinyl-CoA:3-oxoacid CoA transferase), ketone bodies represent a more efficient fuel than glucose and fatty acids [3], thereby enabling the muscle tissue to produce more work for a given energy cost [4]. Owing to the low carbohydrate availability induced by this type of diet, KDs induce a metabolic switch towards a greater reliance on fatty acids, which are required for the production of ketone bodies. Indeed, strong evidence supports the effectiveness of KDs for increasing fat oxidation rates during exercise [5,6,7].

Low-carbohydrate diets, and particularly KDs, have been proposed as beneficial nutritional strategies—at least in the short term—for inducing weight loss and improving cardiometabolic health in both healthy and clinical populations [8,9,10]. The popularity of KDs has also grown considerably among endurance athletes. The reduced reliance on glycogen stores along with increased fat oxidation rates when exercising at submaximal intensities could indeed benefit performance in long-duration events [11,12], although the evidence is mixed [13]. Due to their purported benefits on body composition [2], KDs are also growing in popularity among strength-trained individuals, and indeed these diets have been proposed as an option for some athletes. These include individuals participating in weight-category sports (e.g., combat athletes) or in events where a high ratio of muscle strength relative to body mass is required for success (e.g., jumpers), as well as bodybuilders aiming at minimizing body fat without losing muscle mass during the so-called ‘cutting phase’ [2]. However, controversy exists as to the actual effects of KD on body composition and performance in strength-trained individuals [14,15].

The present narrative review aimed to summarize current evidence on the effects of KD on resistance training-induced changes in body composition and performance, as well as to discuss potential research gaps on this topic. For this purpose, two authors (PLV and ACG) independently conducted a systematic search in PubMed and Web of Science until 2nd August 2021 using the term ‘ketogenic diet’, along with others including ‘athlete’, ‘strength’, ‘power’, ‘force’, ‘training’, ‘trained’, or ‘exercise’. Studies were first screened by title and abstract and the full text of those studies that seemed to meet the inclusion criteria were assessed. We focused mainly on randomized controlled trials conducted with healthy individuals performing strength training and assessing the effects of a KD (>2 weeks, with <10% of total energy intake coming from carbohydrate intake or daily total carbohydrate intake <50 g) compared with a non-KDs. The primary outcome variables were muscle strength or power-related measures and body composition (total body, fat, and muscle mass, respectively). A flow chart of the systematic search is available as Appendix A, and the randomized controlled trials that met the inclusion criteria are summarized in Table 1.

## 2. Effects of Combining Ketogenic Diets with Resistance Training on Body Composition in Trained Individuals

Growing evidence supports the effectiveness of KDs for promoting weight loss in the general population [8]. Although the biological mechanisms explaining this effect remain debatable [16], one potential factor is the higher satiating and thermic effect of proteins [17,18,19], the consumption of which is sometimes increased in KDs. Other proposed mechanisms are the appetite suppressant effects of ketosis [20,21] or the greater rate of fat oxidation coupled with an increased resting energy expenditure as reported in some studies [22,23]. Of note, because glycogen is stored in human cells along with three to four parts of water, the glycogen-depleting effect of KDs could be associated with a further reduction in total body mass [24] (for a graphical summary of these mechanisms, see Figure 1).

Evidence on the effects of KDs on total body and fat mass loss in strength-trained individuals is promising. A recent systematic review and meta-analysis including 13 trials concluded that KDs (5–50 g/d of carbohydrate for 3 to 12 weeks) are effective for reducing total body (−3.7 kg on average) and fat mass (−2.2 kg) compared with non-KDs [15]. Nonetheless, KDs might also contribute to a loss of muscle mass or at least impair resistance training-induced hypertrophy. Indeed, the abovementioned meta-analysis concluded that KDs reduce fat-free mass (−1.3 kg) compared with non-KDs [15].

A summary of relevant randomized controlled trials assessing the effects of KDs on body composition in strength-trained individuals is shown in Table 1. Different studies support the effect of KDs vs. non-KDs in the reduction of both total body and fat mass [25,26,27,28,29,30]. However, some detrimental effects have been reported on muscle mass when KDs are combined with resistance exercise interventions [25,26,27,28,31]. A randomized controlled trial conducted with Olympic-class weightlifters reported that a 3-month *ad libitum* KD resulted in a total loss of body mass above 3 kg compared with a group maintaining their usual diet (~45% carbohydrate) [26]. However, ~77% (2.3 kg) of the weight lost by the KD group was attributed to the muscle component [26]. Another randomized controlled trial conducted with bodybuilders who followed an 8-week energy-balanced KD found a significant reduction in total body and fat mass, respectively, with these changes not reported for a group ingesting an isocaloric westernized diet (~55% carbohydrate) [25]. Of note, only bodybuilders in the westernized diet showed an increase in muscle mass [25]. Two recent studies by the same group of researchers analyzed the effects of an 8-week KD in strength-trained men and women [27,28]. The diets were designed to induce no energy deficit and even to produce a moderate energy surplus through a total energy intake of 39 kcal/kg body mass in men [27] and 40–45 kcal/kg fat-free mass in women [28]. Overall, the KD resulted not only in a lower energy intake (~1710 vs. 1979 kcal and 40.1 vs. 45.5 kcal/kg fat-free mass per day for the KD and the non-KD diet, respectively, in the study conducted in women) compared with the control non-KD group (>55% kcal from carbohydrate), but also in a reduced fat mass (by ~0.8 and 1.1 kg for men and women, respectively). However, whereas the control groups showed a trend to increase fat-free mass (~by 1.3 and 0.7 kg for men and women, respectively), participants on the KD showed no changes [27,28]. Recently, Kysel et al. [31] found a comparable reduction in body mass in healthy young resistance-trained men who combined resistance and aerobic training with (i) a hypocaloric cyclical KD (i.e., 500-kcal energy deficit alternating a phase of KD for 5 days followed by 2 days of carbohydrate reintroduction) or (ii) a hypocaloric non-KD diet, for 8 weeks. Although no significant between-group differences were found at post-intervention (KD −4.6 kg vs. non-KD, 4.5 kg), only participants assigned to the KD treatment showed a significant reduction in muscle mass (KD −1.8 kg vs. non-KD −0.4 kg). On the other hand, Rhyu et al. [32] reported similar losses in body mass (including both fat and muscle components) in high school taekwondo athletes following a 3-week period of hypocaloric KD or non-KD diet intervention. Additionally, Skemp et al. [30] observed larger reductions of body mass and fat mass in resistance-trained women who followed a 4-week hypocaloric, KD or non-KD. Both treatments induced similar detrimental effects on muscle mass. More recently, Vidic et al. [33] reported that both a hypocaloric KD and a hypocaloric low-carbohydrate but non-KD (with carbohydrates accounting for 5 and 15% of total energy intake, respectively, both inducing a total energy deficit of ~600 kcal/d) induced similar losses in total body (−6.1 and −5.3 kg, respectively) and fat mass (−4.3 and −3.5 kg) in strength-trained individuals, although a decrease in muscle mass (−1.8 and −1.5 kg) was also reported with both diets.
nutrients-13-03083-t001_Table 1Table 1Summary of randomized controlled trials that have assessed the effects of ketogenic diets (KD) on body composition or performance in healthy strength-trained individuals.StudyParticipantsDuration of InterventionKD CDMain Findings Body CompositionPerformanceKysel et al. [31] 25 strength-trained men 8 weeksControlled energy intake (500 kcal of energy deficit)Fat: NS CHO: NS (<30 g) Protein: 1.6 g/kgIncluding 2 days of CHO re-introduction (CHO 70%) each 5 days.Controlled energy intake (500 kcal of energy deficit)Fat: 30% CHO: 55%Protein: 15%↓ Muscle mass and water content only with KD.Similar ↓ in body mass and fat mass with both diets. ↑ maximal muscle strength (lateral pull down and leg press) only with CD.↑ cardiorespiratory fitness (peak oxygen uptake and peak workload) only with CD. Paoli et al. [25]19 competitive male bodybuilders8 weeksControlled energy intake (isocaloric)Fat: 68% CHO: 5% (44 g)Protein: 25% (216 g, 2.5 g/kg)Controlled energy intake (isocaloric)Fat: 20%CHO: 55%Protein: 25% (223 g, 2.5 g/kg)↓ Body mass only with KD.↓ Fat mass only with KD.↑ Muscle mass only with CD. ↑ strength (1RM in squat and bench press) similarly in both groups.Rhyu et al. [32] 20 young (15–18 years) Taekwondo athletes3 weeksControlled energy intake (hypocaloric, 75% of estimated energy intake)Fat: 55% CHO: 4.3% (22 g)Protein: 40.7% Controlled energy intake (hypocaloric, 75% of estimated energy intake)Fat: 30% CHO: 40%Protein: 30%Similar ↓ in total body mass, fat mass and muscle mass for both groups.↑ in 2000-m running trial performance and Wingate test performance (fatigue index) with KD.Similar ↓ in peak and mean power on the Wingate test with both diets. Similar ↑ in back muscle strength and in the number of sit-ups with both diets.No changes in performance in the remaining outcomes.Skemp et al. [30]20 strength-trained women4 weeksAd libitumFat: 70%CHO: 10% Protein: 20%Ad libitum (normal standard diet)Fat: NSCHO: NSProtein: NS↓ Body and fat mass with KD vs. CD.Similar ↓ of muscle mass with both diets.N/AGreene et al. [26]14 elite competitive lifting athletes (5 female)12 weeksAd libitumFat: 70%CHO: 8% (39 g)Protein: 23% (120 g, 1.6 g/kg)Ad libitumFat: 33%CHO: 45% (223 g)Protein: 22% (120 g, 1.5 g/kg)↓ of both body mass and muscle mass after KD vs. CD.No differences in performance.Wilson et al. [29]25 strength-trained men11 weeks (10 weeks of KD + 1 week of CHO re-introduction)Controlled energy intake (isocaloric)Fat: 75%CHO: 5% (31 g)Protein: 20% (134 g, 1.7 g/kg)Followed by a week of CHO reintroduction (increasing from 1 to 3 g/kg of CHO during the last week) Controlled energy intake (isocaloric)Fat: 25%CHO: 55% (318 g)Protein: 20% (132 g, 1.7 g/kg)↓ Fat mass with KD vs. CD Similar ↑ in muscle mass and thickness, but greater ↑ with KD after CHO reintroduction.Similar performance in 1RM with CD and KD, although only the former increased peak power in the Wingate test.Vargas et al. [27]24 strength-trained men8 weeksControlled energy intake (moderate energy surplus, 39 kcal/kg)Fat: 70%CHO: <10% (42 g)Protein: 20% (2.0 g/kg)Controlled energy intake (moderate energy surplus, 39 kcal/kg)Fat: 25%CHO: 55% Protein: 20% (2.0 g/kg)↓ Fat mass with KD (no significant interaction effect) ↑ Muscle mass and body mass only with CD. N/AVargas et al. [28]21 strength-trained women8 weeksControlled energy intake (moderate energy surplus, 40–45 kcal/kg FFM)Fat: 64%CHO: 9% (30–40 g)Protein: 27% (115 g, >1.7 g/kg)Controlled energy intake (moderate energy surplus, 40–45 kcal/kg FFM). Significantly higher energy intake than KD. Fat: 23%CHO: 57% (282 g)Protein: 20% (>1.7 g/kg)↓ Body mass and fat mass with KD vs. KD.↑ Muscle mass with CD vs. KD.↑ Bench press and squat performance (1RM) with CD vs. KD. Similar improvements in CMJ performance.Vidic et al. [33]20 strength-trained men8 weeksControlled energy intake (hypocaloric)Fat: 75%CHO: 5% (27 g)Protein: 20% (108 g, 1.2 g/kg)Controlled energy intake (hypocaloric, non-ketogenic)Fat: 65%CHO: 15% (82 g)Protein: 20% (110 g, 1.2 g/kg)Similar ↓ in body mass, fat mass and muscle mass with KD and CD.No changes in performance (1RM) with any of the diets.↑ and ↓ indicate significant increases and reductions, respectively. Abbreviations: 1RM, one-repetition maximum; CD, control diet; CHO, carbohydrate; CMJ, countermovement jump; FFM, fat-free mass; KD, ketogenic diet; NS, not specified.


Some non-randomized interventional studies have also assessed the effects of KD on body composition. In previously untrained overweight women who started a 10-week resistance training program, Jabekk et al. reported a higher total body (−5.6 kg) and fat mass (−5.6 kg) loss in participants following a KD compared to those maintaining their usual eating pattern [34]. However, only those participants who maintained their habitual diet showed significant increases in muscle mass (1.6 kg) [34]. A non-randomized controlled trial conducted in CrossFit athletes who followed a 3-month KD intervention found a ~3 kg and −2.5 kg reduction in total body mass and fat mass, respectively, with no significant changes reported in those who chose to maintain their usual diet. Of note, the body mass loss of the KD groups was also accompanied by a non-significant reduction (−0.4 kg) in lower-limb muscle mass [35]. A non-randomized cross-over study in 8 elite artist gymnasts found that, contrary to a usual western diet, 30 days of KD led to a reduction in total body (−1.6 kg) and fat mass (−1.9 kg), while muscle mass remained constant (non-significant reduction of 1.1 kg) [36]. A recent study tested the hypothesis that providing an exogenous ketone supplement (ketone salts) combined with a hypocaloric (75% of estimated energy needs) KD might help to preserve muscle mass. Although a trend towards a lower nitrogen excretion was found—which could have potential implications for muscle mass preservation in the long term—no actual benefits on muscle mass or overall body composition were observed [37]. 

The bulk of the evidence that is currently available therefore suggests that combining 8 to 12 weeks of KD with resistance training can a favor fat mass reduction in healthy and trained individuals, although muscle mass accretion might be also compromised, at least partly. 

### Mechanisms Underlying the Detrimental Effects of Ketogenic Diets on Muscle Mass

Several mechanisms have been proposed to explain the potential detrimental effects of KD on muscle mass [38,39] (Figure 2). Due to the hydrophilic properties of the surface of glycogen granules [40], KD-induced glycogen reductions may explain the loss of muscle mass [24]. Indeed, to the best of our knowledge, only one randomized controlled trial to date has reported superior hypertrophic effects in resistance-trained participants following a KD compared with a non-KD [29]. Wilson et al. found similar gains in muscle mass with an energy-balanced KD or an isocaloric traditional (~55% kcal from carbohydrates) western diet (increase in muscle mass of 2.4 vs 4.4%, respectively) after a 10-week intervention period in which participants in both groups consumed ~1.7 g/kg of protein [29]. However, when reintroducing carbohydrates for one week (in weeks 10–11 of the study), participants on the KD increased their muscle mass to a greater extent than those on the western diet [29]. These findings might support the importance of avoiding a prolonged glycogen depletion status and the potential benefits of reintroducing carbohydrates in order to preserve or even increase muscle mass in athletes following a KD. They might also support the occurrence of the so-called ‘sarcoplasmic hypertrophy’, that is, an increase in muscle volume caused by sarcoplasmic expansion, which in turn is due to a greater water content (‘retention’) because of the hydrophilic nature of the surface of glycogen granules, rather than by ‘sarcomeric hypertrophy’ (i.e., an actual increase in myofibril protein content) [41].

It must be noted that besides the potentially ‘confounding’ effects of low glycogen stores on ‘real’ (i.e., protein content) muscle mass, low carbohydrate availability might also attenuate resistance training-induced adaptations through a suppression of anabolic pathways. By virtue of the reduced carbohydrate intake, KD can indeed lead to a reduction in insulin levels [25,33], with this hormone having been in turn reported to stimulate muscle protein synthesis—at least when combined with concomitant increases in amino acid availability—and to reduce muscle protein breakdown [42,43,44]. Preclinical evidence suggests that compared with isocaloric control diets, KD might promote AMP-activated protein kinase (AMPK) phosphorylation [45], which can inhibit anabolic pathways such as kinase B protein (Akt)/mechanistic target of rapamycin (mTOR) [46]. Additionally, preclinical evidence in isolated mouse muscle suggests that ketosis (e.g., presence of BHB) can diminish Akt phosphorylation [47], thereby impairing anabolic responses. Conversely, other studies in non-athletic populations have reported that ketosis might induce anticatabolic effects [48] by increasing circulating levels of ketone bodies (induced through oral or intravenous administration of ketone bodies) that in turn help to preserve muscle mass and maintain nitrogen balance during conditions of energy deficit [49,50]. In the same line, ketone bodies have been shown to exert anti-inflammatory [51] and antioxidant effects [52], which could have potentially beneficial effects against muscle wasting [53]. In fact, lower levels of inflammatory and oxidative stress markers have been reported in athletes undergoing a KD compared with those following a conventional western diet [32,54]. Further research is therefore needed to confirm the role of ketosis on muscle anabolism, particularly on healthy and trained individuals.

The appetite-suppressant effects of ketosis [20,21] might also play a role on the muscle mass loss observed with KD. Thus, ad libitum KD might result in a reduced caloric intake compared with conventional western diets, which can in turn have detrimental consequences on muscle protein synthesis and muscle mass accretion [55,56]. Moreover, chronic low energy availability [57] and acute severe energy restriction—which is commonly experienced by athletes who seek to rapidly lose body mass before competition—have been reported to exert detrimental effects on the hormonal environment, notably reduced levels not only of circulating total testosterone, but also of thyroid-stimulating hormones [58], with thyroid hormone signaling playing an important role in muscle homeostasis and repair [59]. In addition, although caloric restriction might have no effects on anabolic hormones such as growth hormone (GH) or insulin-like growth factor 1 (IGF-1) [60,61], it can lead to a reduction in sex hormones such as testosterone [62]. However, controversy exists as to whether energy-balanced KD can also affect testosterone levels [63]. Indeed, because lipids and derivatives (particularly cholesterol) are the substrate for the biosynthesis of androgens, high-fat diets (e.g., KD) might be potentially beneficial for promoting testosterone synthesis, at least when sufficient energy is provided [63]. Nonetheless, the evidence is mixed. In healthy moderately active individuals, Volek et al. reported no changes in testosterone concentration after 8 weeks of a high-fat diet, although this study did not include a control group and participants performed no strength training [64]. Regarding strength-trained individuals, four recent studies [25,29,33,65] have reported mixed results. Paoli et al. found a decrease in the concentration of anabolic hormones (testosterone and IGF-1) after 2 months of an energy-balanced KD (45 kcal/kg muscle mass) in bodybuilders [25]. Wilson et al. reported no differences in free testosterone along with a significantly higher total testosterone concentration in resistance-trained college athletes after a 10-week KD compared with a western isocaloric diet. However, these differences between diets were found after one week of carbohydrate reintroduction [29]. Vidic et al. observed similar increases in testosterone in strength-trained individuals exposed to a hypocaloric KD and a hypocaloric non-KD [33]. Moreover, Michalczyk et al. [65] reported an increase in the levels of GH and testosterone in male basketball players after 4 weeks of a low-carbohydrate diet (10% of total energy intake) compared with a previous period in which they consumed the same energy intake through their conventional diet. Of note, after a week of carbohydrate reintroduction, testosterone levels remained above baseline levels, but those of GH declined against those observed with the conventional diet [65]. Thus, current evidence shows no consistent effects of KD on the hormonal anabolic environment, although there are mixed and scarce results as well as some confounding factors (e.g., differences between studies in total caloric or protein intake, or in carbohydrate reintroduction), all of which might hinder drawing conclusions.

## 3. Effects of Combining Ketogenic Diets with Resistance Training on Strength and Power Performance in Trained Individuals

Given the promising results of KD on body composition, further performance benefits of KD could be hypothesized at least in those sports in which body mass is a key determinant (e.g., weight-category sports or those involving actions performed with the own body mass such as jumps) [2]. In turn, the detrimental effects of KD on muscle mass could result in an impaired muscular function, especially when muscle performance is expressed in absolute values (e.g., total kg lifted, or total wattage produced) instead of relative to body mass (e.g., total kg lifted/kg or watts/kg). For this reason, it is important to explore whether muscle strength is improved or at least maintained during KD.

Conflicting results exist regarding the effects of KD on muscle strength or other related performance measures (e.g., power output). Reflecting this controversy, a recent systematic review assessed seven studies that had analyzed the effects of KD on strength or power measures on 16 performance outcomes studied (mainly muscle strength (one-repetition maximum, 1RM in different exercises, such as jump performance and sprint power output) [14]. Only two reported a significantly beneficial effect of KD, whereas 11 found no effects and 3 observed an impaired performance after a KD [14]. Nonetheless, it must be noted that the two performance measures in which benefits were observed corresponded to two cycling tests (6-s sprint and 3-min critical power test) that were implemented in the same study after a 100-km trial [66]. As such, the performance measure in question might have been more indicative of muscle endurance than of muscle power capacity.

Table 1 summarizes randomized controlled trials that have assessed the effects of KD on muscle strength- or power-related outcomes in strength-trained individuals. Overall, no effects of KD on strength or power-related performance have been reported. Indeed, except for the study by Rhyu et al., which reported greater benefits with a KD compared with a non-KD on a 2000-m running trial and on the ‘anaerobic’ fatigue index assessed by the Wingate test [32], the remainder of randomized controlled trials reported no beneficial effects of KD on performance or even detrimental effects. It must be noted, however, that some studies have found no changes in performance outcomes despite reporting losses in total body (and even muscle) mass, which could be considered beneficial in some specific situations [26,35]. For instance, Vidic et al. reported no changes in squat and bench press 1RM after an 8-week hypocaloric KD that induced an average body mass loss of 6.1 kg [33]. Greene et al. found no variations in 1RM strength on different exercises despite a total body mass loss of 3.2 kg compared with a control (participants’ usual) diet [26]. Kephart et al. observed that participants who self-selected to follow a KD during three months maintained their 1RM on the squat and power clean exercises despite a total body mass loss of 3 kg [35]. Similarly, Paoli et al. reported that 30 days of KD did not have a significant impact on performance (jump height, grip chins and push-ups) despite a body mass loss of 1.6 kg [36]. Nonetheless, other studies have found a detrimental effect of KDs on performance—or at least lower benefits—compared to a traditional western diet. Vargas et al. reported greater improvements in performance (jump height and 1RM bench press) after a non-KD compared with an 8-week KD [28]. In the study by Wilson et al., those participants who followed a non-KD over 10 weeks increased their peak power output on a Wingate test, whereas those who followed a KD did not, albeit with no significant differences in the pre-post change between conditions [29]. Kysel et al. reported larger benefits on maximal strength (1RM on the bench press and lateral pull-down exercises) and on some markers of cardiorespiratory fitness (peak oxygen uptake and peak workload during an incremental test) in individuals who followed a hypocaloric non-KD compared with those who followed a hypocaloric cyclical KD [31]. In a non-controlled study by Urbain et al., participants who followed a KD for 6 weeks showed an impairment of peak power during an incremental cycling test (−4.1%), although handgrip strength increased slightly (+2.5%) [67]. Similarly, Fleming et al. observed that a 6-week high-fat diet (61% fat, 8% carbohydrate) resulted in a reduced peak and mean power during a Wingate test compared with a control diet [68]. More recently, a study conducted in CrossFit athletes revealed that a 4-week KD induced no beneficial effects on CrossFit-specific performance (assessed through a workout including jumps, push presses and rowing, among other exercises) and even resulted in an impaired cardiorespiratory fitness (lower peak oxygen uptake) in women [69]. 

In summary, although some studies support the effectiveness of KD for reducing total body, and particularly fat, mass in strength-trained individuals without harming sports performance, there is also evidence for some performance decrements compared to non-KD western diets.

## 4. Perspectives

Evidence on the effects of KD on strength-trained individuals is rapidly growing. Many studies have methodological limitations (e.g., not following a randomized controlled trial design, not monitoring dietary intakes, small sample sizes, short duration of the intervention, or variation in the amount and type of carbohydrates (high or low glycemic index) during the intervention) impeding us in the drawing of evidence-based inferences. In addition, as shown in Table 1, the number of randomized controlled trials conducted in healthy strength-trained individuals is still scarce, and no study has assessed the long-term effects (>12 weeks) of KD in this population.

There is only one study to date reporting muscle mass gains after a KD, with this effect found after reintroducing carbohydrates for one week [29]. Similarly, Michalczyk et al. recently reported that, although following a low-carbohydrate diet (10% of total energy intake) for 4 weeks resulted in an impaired performance during the Wingate test (−11% total work capacity) compared to a conventional diet, after reintroducing carbohydrates for one week (75% of total energy intake), performance recovered to levels similar to those observed with the conventional diet [65]. These findings suggest that carbohydrate reintroduction might be an optimal pre-competition strategy to avoid the potential detrimental consequences of KD on muscle mass and performance in those athletes not concerned about potential increases in body mass. However, another trial that applied a hypocaloric cyclical KD (i.e., by alternating 5 days of KD with 2 days of high carbohydrate intake (70% of total energy intake)) reported lower performance gains and a greater loss of muscle mass compared with a regular non-KD designed to induce the same energy deficit (−500 kcal in both cases) [31]. Further research is therefore warranted to confirm whether including a carbohydrate reintroduction phase might mitigate some of the detrimental consequences of KD. 

The neutral effects of KD on absolute strength/power (e.g., 1RM, maximal power output) might support their potential benefits on muscle strength/power relative to total body mass (e.g., watts/kg), on performance in weight-bearing exercises such as jumps, push-ups, pull-ups, and also for athletes competing in weight-category sports (e.g., combat sports), although the evidence is still controversial and overall not promising [28,36,69]. Further research, including randomized controlled trials, is needed to confirm the actual effects of KD on performance outcomes in which body mass plays an important role.

Although more research is also needed to confirm the exact mechanisms underlying the impairment effect of KD vs. non-KD on resistance training-induced hypertrophy, it might be recommendable to closely monitor the protein intake of KD. As summarized in Table 1, several studies, including those that found a reduced muscle mass with the KD, have provided daily protein intakes ranging from between 1.2 and 2.5 g/kg/day, with similar protein intakes in those individuals who followed a KD or a non-KD. In this regard, protein intakes of 1.6–2.0 g/kg/day have been recommended to maximize resistance training-induced gains in muscle mass and strength [70,71,72]. However, under conditions of energy restriction, higher protein intakes (1.7 to 3.1 g/kg) might be needed to maintain muscle mass [71,73]. Moreover, it has been reported that, compared with an energy-matched high-carbohydrate diet, higher protein intakes might be needed for those people on a low-carbohydrate diet in order to meet protein requirements during post-exercise recovery [74]. Bodybuilders who followed an 8-week KD with a protein intake of 2.5 g/kg/day were able to maintain their muscle mass, although those who followed a non-KD westernized diet increased their muscle mass to a greater extent [25]. Further evidence is therefore needed to confirm whether increasing protein intake through diet or protein (>1.6 g/kg)/ amino acid (e.g., leucine) supplementation can negate the potentially detrimental effects of KD on muscle anabolism. Conversely, given the rapid adaptation of the human body to maximize liver gluconeogenesis under situations of increasing aminoacidemia and low carbohydrate availability, following a KD with high protein intake could stimulate hepatic gluconeogenesis from proteins [75,76], with a subsequent reduction in ketosis. However, the role of dietary protein on gluconeogenesis remains controversial [77]. Indeed, high circulating ketone levels (>1 mmol/L) have been reported even with very low carbohydrate diets coupled with high protein intakes (0 and 30% of the total energy intake, respectively) despite the occurrence of gluconeogenesis [78]. The role of protein intake on the effects of KD should therefore be further addressed. 

Future studies should also determine a range of effective nutritional ketosis. Wilson et al. [29] reported circulating ketone levels consistently surpassing >0.5 mmol/L and reaching ≥1 mmol/L after 3 weeks of isocaloric KD. Similarly, Vidic et al. [33] observed that those individuals following a hypocaloric KD presented steady blood ketone levels of 1–2 mmol/L, whereas those following a hypocaloric LCD but non-KD had ketone concentrations of 0.1–0.2 mmol/L. Other studies have confirmed the presence of urinary ketones using reagent strips, with some of them removing from the study those individuals who did not show positive ketosis [27,28] and others just confirming that participants were under nutritional ketosis for most of the days studied (range 69–100%) [67]. In turn, Greene et al. [26] and Fleming et al. [68] observed an average ketone concentration <0.5 mmol/L (0.4 and 0.3 mmol/L, respectively) after a KD, despite keeping carbohydrate intake <10% of total energy intake (or <50 g/d), which might reflect that some individuals did not adhere to the dietary recommendations or did not attain ketosis. Future studies should confirm whether higher levels of circulating ketones (e.g., >1.0 instead of 0.5 mmol/L) can maximize KD benefits. Preliminary evidence combining a KD intervention with an exogenous ketone supplement vs. a KD alone failed to find any additional benefit in spite of reaching higher levels of ketosis, particularly during the first few weeks [37]. 

Another potentially confounding factor might be the energy intake associated with KD. Evidence overall suggests that KDs are more effective than western diets for promoting loss of total body and fat mass. Thus, studies have reported that ad libitum KD results in a greater loss of total body or fat mass than an ad libitum western diet [26,30]. In the same line, studies comparing energy-balanced KD with isocaloric western diets also show superior benefits of the former on total body and fat mass reduction [25,27,29]. However, studies comparing hypocaloric KD with western diets or non-KD low-carbohydrate diets resulting in the same energy deficit have found a similar effect on total body/fat mass [31,32,33]. More controversy exists, however, on how energy intake might affect the effects of KD on muscle mass. Thus, a study analyzing the effect of an energy-balanced KD reported that it was as effective as a non-KD western diet for increasing muscle mass [29]. In turn, other authors have found that an energy-balanced KD is less effective than an isocaloric western diet for improving muscle mass [25], and others have reported losses in muscle mass with ad libitum and hypocaloric KD, this loss of muscle mass being greater than that observed with an ad libitum western diet [26,33]. Further research analyzing the effects of KDs with different energetic conditions (energy-balanced vs. hypocaloric vs. hypercaloric) is needed to draw definite conclusions on the influence of energy balance on the effects of KD, as well as to compare the actual effectiveness for promoting fat loss of hypocaloric KD and conventional diets resulting in the same energy intake.

Finally, a major concern with KD is their eventual long-term safety [79]. KDs have been overall reported to be safe and to improve different cardiovascular disease risk factors such as obesity and glucose metabolism, although the long-term sustainability of KD-induced benefits remains unclear [80,81]. Moreover, a great proportion of individuals starting KDs report several symptoms (known as ‘keto flu’) during the first weeks, including headache, fatigue, nausea, dizziness and gastrointestinal discomfort [82]. There is also evidence of increased levels of low-density lipoprotein cholesterol and apo-B-containing lipoprotein with this type of diet [83]. It should be kept in mind that, as with any diet (including low-fat diets), the quality of the nutrients ingested (e.g., ultra-processed vs unprocessed or minimally processed foods, refined vs unrefined carbohydrates and saturated vs unsaturated fats) should be a primary focus [80]. In this regard, given that KDs are typically characterized by a high intake of saturated fats or animal-based foods and also by a low fiber intake, which could be detrimental for cardiovascular health, inclusion of polyunsaturated fats (as found in avocado, nuts, coconut or olive oil) and plant-based foods that are also rich in proteins (e.g., tofu, pea, tempeh, seitan) might be recommendable [84,85]. 

## 5. Conclusions

Evidence overall supports the effectiveness, at least in the short term, as no study has yet assessed the long-term effects of these diets, of KDs for reducing total body and fat mass in strength-trained individuals compared with non-KDs. Nonetheless, further research is needed to confirm the superiority of hypocaloric KDs over non-KDs with the same energy intake. Conversely, KDs might impair resistance training-induced gains on muscle mass and performance, particularly when expressed in absolute values (e.g., total kg lifted, watts). Further evidence is needed regarding the long-term safety of these diets. Caution should be therefore taken when maintaining a KD in the long term or when increases in muscle mass and performance are sought.

## Figures and Tables

**Figure 1 nutrients-13-03083-f001:**
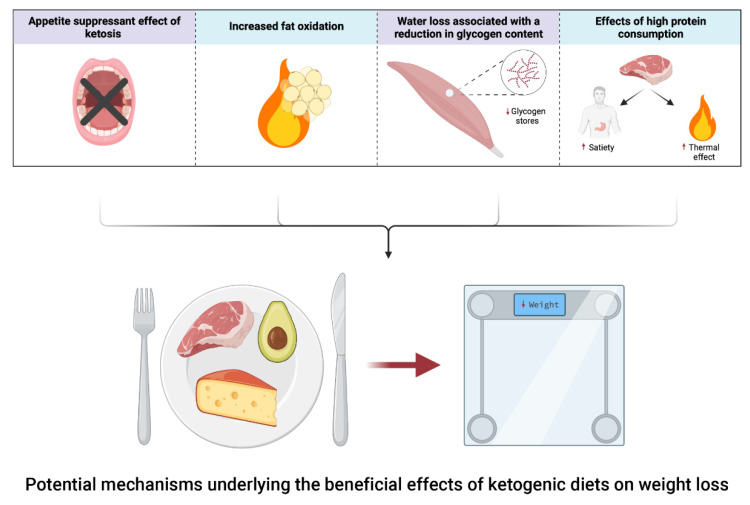
Summary of some potential mechanisms underlying the beneficial effects of ketogenic diets on weight loss.

**Figure 2 nutrients-13-03083-f002:**
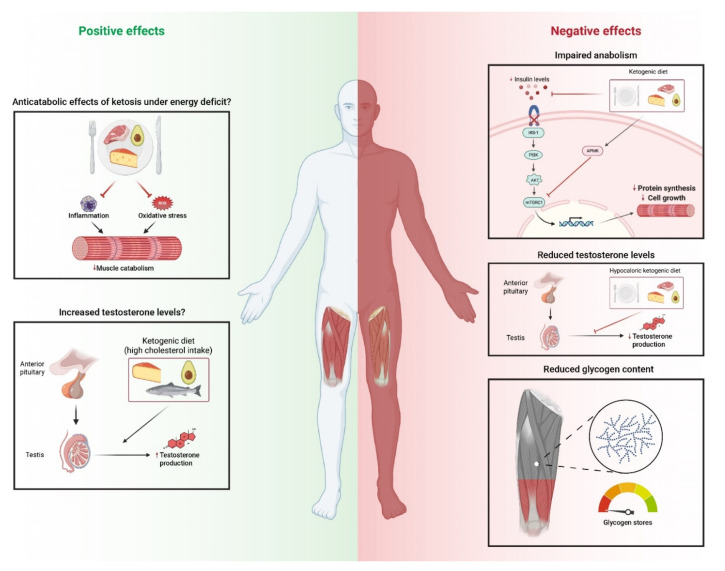
Summary of some potential mechanisms underlying the positive and negative effects of ketogenic diets on muscle mass.

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
