# Peer review of "Effects of Combining a Ketogenic Diet with Resistance Training on Body Composition, Strength, and Mechanical Power in Trained Individuals: A Narrative Review"

_nutrients, 2021, doi:10.3390/nu13093083_

Round 1

Reviewer 1 Report

This narrative review aimed to summarize current evidence on the effects of Ketogenic Diet on resistance training-induced gains in body composition and performance.

Despite these premises,  the authors fail to provide a comprehensive insight providing only few information that are easily retrievable with a simple search on PubMed.   
The method for choosing the articles included in the review is not clear. The authors emphasize only the benefits of the ketogenic diet and selected predominantly papers whose control group are subject to free or low-calorie diets. It is clear that a strongly hypocaloric diet, which not surprisingly is maintained for short periods of time, has major advantages in terms of fat mass reduction over isocaloric or mildly hypocaloric diets. 

Although it is a narrative review it would be useful to Include in the paper the methods used to search for articles and a flow-chart (PRISMA).

The authors should completely revise the paper:
1. the title and abstract are misleading and do not immediately clarify the inconsistencies and few benefits of ketogenic diets in sports
2. Many papers show conflicting results regarding the effects of a KD on sporting performance, this should be highlighted.
3. The absence of long-term studies is essential and should be emphasized. We suggest to athletes a type of diet that gives good results in the short term (although probably you could achieve the same goals of reducing fat mass without penalizing carbohydrates) but unknown in the long term. 
4. It is appropriate to differentiate low-protein or high-protein ketogenic diets. A diet so strongly restrictive on carbohydrates educates the subject to consider more healthy foods rich in proteins and fats. Unless foods rich in plant-based proteins and mono- or polyunsaturated fats are used, this information puts the subject's health at risk. Is it right to propose a diet that gives small advantages on fat mass but big disadvantages for health and deleterious effects on the environmental footprint? 

Author Response

Dear Reviewer,

Thank you very much for the thorough review of our work and for their constructive comments. We have revised the manuscript and believe it will make a positive and required contribution to the literature in the field.

Responses

All points raised and suggestions have been seriously considered and responded on point-by-point basis. Please see below.

This narrative review aimed to summarize current evidence on the effects of Ketogenic Diet on resistance training-induced gains in body composition and performance. Despite these premises, the authors fail to provide a comprehensive insight providing only few information that are easily retrievable with a simple search on PubMed.   
The method for choosing the articles included in the review is not clear. The authors emphasize only the benefits of the ketogenic diet and selected predominantly papers whose control group are subject to free or low-calorie diets. It is clear that a strongly hypocaloric diet, which not surprisingly is maintained for short periods of time, has major advantages in terms of fat mass reduction over isocaloric or mildly hypocaloric diets. Although it is a narrative review it would be useful to Include in the paper the methods used to search for articles and a flow-chart (PRISMA).

We appreciate the Reviewer’s comments. Following the Reviewer’s suggestion we have now performed a systematic search in two main databases (PubMed and Web of Science) in order to find all the published randomized controlled trials (RCTs) that have assessed the effects of ketogenic diets in the body composition or performance of strength-trained individuals (please see revised text and new Supplementary Figure 1 for the flowchart of the systematic search). As a result, 3 new RCTs have now been added to our manuscript (including Table 1) and discussed accordingly. Of note, regardless of whether the control group was given a hypocaloric or an ad libitum diet, all the RCTs assessing the aforementioned question have been included and discussed along the paper. Indeed, as summarized in Table 1, we have included several RCTs in which both the ketogenic diet group and the control group had the same energy intake. We have also added a paragraph in the Section 4 (“Perspectives”) in which we have elaborated more on our discussion on the question of the energy intake associated with the diets (i.e., ad libitum, hypocaloric, or energy-balanced). Also, please note that we have tried not to emphasize the benefits of ketogenic diets; indeed, throughout the manuscript –including the abstract and conclusions section– we mention that these diets might actually have detrimental effects on muscle mass and performance.

The authors should completely revise the paper:
1. the title and abstract are misleading and do not immediately clarify the inconsistencies and few benefits of ketogenic diets in sports.

We have now modified our title to better describe the main aspects and contributions of our paper.

The title now reads as follows:

Effects of combining ketogenic diets with resistance training on body composition, strength, and mechanical power in trained individuals: a narrative review

We have also rephrased the Abstract section to emphasize the lack of long-term studies and the controversy regarding the effects of ketogenic diets on body composition and muscle function (strength and mechanical power). Additionally, in line with the Reviewer’s suggestion we have also rephrased some of the headings in Sections 2 and 3.

  1. Many papers show conflicting results regarding the effects of a KD on sporting performance, this should be highlighted.

Thanks again for helping us to improve our review. We have reviewed the main characteristics of all included papers as well as the two main analyzed outcomes, that is, intervention-induced changes in (i) body composition and (ii) muscle strength/mechanical power. In line with the above comments, we have also integrated several sentences emphasizing some controversies between studies that result from combining KD with resistance training in trained individuals.

The following sentence has been modified in the Abstract

Although long-term studies (>12 weeks) are lacking, growing evidence supports the effective-ness of an ad libitum and energy-balanced KD for reducing total body and fat mass, at least in the short term. However, no – or negligible – benefits on body composition have been observed when comparing hypocaloric KD with conventional diets resulting in the same energy deficit. Moreover, some studies suggest that KD might impair resistance training-induced muscle hypertrophy, sometimes with concomitant decrements in muscle performance, at least when ex-pressed in absolute units and not relative to total body mass (e.g., one-repetition maximum).

Please see also several new or modified sentences into Sections 2 and 3.

  1. The absence of long-term studies is essential and should be emphasized. We suggest to athletes a type of diet that gives good results in the short term (although probably you could achieve the same goals of reducing fat mass without penalizing carbohydrates) but unknown in the long term. 

We agree with the Reviewer on the importance of this issue, which was already highlighted and discussed. In any case, we have now mentioned this issue also in the Abstract as well as in Sections 4 (‘Perspectives’) and 5 (‘Conclusions’).

  1. It is appropriate to differentiate low-protein or high-protein ketogenic diets. A diet so strongly restrictive on carbohydrates educates the subject to consider more healthy foods rich in proteins and fats. Unless foods rich in plant-based proteins and mono- or polyunsaturated fats are used, this information puts the subject's health at risk. Is it right to propose a diet that gives small advantages on fat mass but big disadvantages for health and deleterious effects on the environmental footprint? 

Thanks for this valuable comment. We have now included a new paragraph at the end of Section 4 (‘Perspectives’) where we discuss the safety aspects of ketogenic diets and highlight the importance of prioritizing polyunsaturated fats and plant-based foods. In addition, the lack of evidence on the long-term safety of ketogenic diets is now mentioned in the Abstract and in the Conclusions sections.

On behalf of all authors, many thanks for this insightful review.

Reviewer 2 Report

The work is well structured and covers a "hot" and actual topic.

line 61: you could mention some work concerning making weight (for example Cannataro R, Cione E, Gallelli L, Marzullo N, Bonilla DA. Acute Effects of Supervised Making Weight on Health Markers, Hormones and Body Composition in Muay Thai Fighters. Sports (Basel). 2020 Oct 16; 8 (10): 137. Doi: 10.3390 / sports8100137) also underlining an effect on thyroid hormones that obviously would have an effect on muscle mass

line 73: it should be emphasized the anti-inflammatory effect of the ketogenic diet (for example Dupuis N, Curatolo N, Benoist JF, Auvin S. Ketogenic diet exhibits anti-inflammatory properties. Epilepsia. 2015 Jul;56(7):e95-8. doi: 10.1111/epi.13038), which is very important in particular in obese subjects, suffering from low-grade inflammation, as well as the positive action on the antioxidant state (for example Cannataro R, Caroleo MC, Fazio A, La Torre C, Plastina P, Gallelli L, Lauria G, Cione E. Ketogenic Diet and microRNAs Linked to Antioxidant Biochemical Homeostasis. Antioxidants (Basel). 2019 Aug 2;8(8):269. doi: 10.3390/antiox8080269), perhaps to be added also in the elegant figure 1

It was pointed out that the decrease in muscle mass is at least in part due to the depletion of glycogen and consequent loss of water, which is however recovered by reintroducing carbohydrates. 

I would also underline that if in a low-calorie regime the anorectic effect is an advantage in terms of research of hypertrophy, it is a disadvantage, so even in ad libitum regimes you are not able to take enough calories

As for the mTOR pathway, a leucine supplement could be useful, as it is a direct and insulin-dependent activator of the pathway.

Finally, while probably not the ideal scheme to promote hypertrophy, it could be important to establish an effective protocol (perhaps alternating carbohydrate refills) for long-term periods of ketogenic dictated by pathological states such as epilepsy, migraine, or lipedema.

Author Response

Dear Reviewer,

Thank you very much for the thorough review of our work and for their constructive comments. We have revised the manuscript and believe it will make a positive and required contribution to the literature in the field

Responses

All points raised and suggestions have been seriously considered and responded on point-by-point basis.

The work is well structured and covers a "hot" and actual topic.

Comments are much appreciated.

line 61: you could mention some work concerning making weight (for example Cannataro R, Cione E, Gallelli L, Marzullo N, Bonilla DA. Acute Effects of Supervised Making Weight on Health Markers, Hormones and Body Composition in Muay Thai Fighters. Sports (Basel). 2020 Oct 16; 8 (10): 137. Doi: 10.3390 / sports8100137) also underlining an effect on thyroid hormones that obviously would have an effect on muscle mass

Thank you very much for the suggestion. We have now stated that acute severe energy restriction (such as that experienced by athletes ‘making weight’) can reduce the levels of thyroid hormones, which would have a negative impact on muscle mass. The reference proposed by the Reviewer has now been included.

line 73: it should be emphasized the anti-inflammatory effect of the ketogenic diet (for example Dupuis N, Curatolo N, Benoist JF, Auvin S. Ketogenic diet exhibits anti-inflammatory properties. Epilepsia. 2015 Jul;56(7):e95-8. doi: 10.1111/epi.13038), which is very important in particular in obese subjects, suffering from low-grade inflammation, as well as the positive action on the antioxidant state (for example Cannataro R, Caroleo MC, Fazio A, La Torre C, Plastina P, Gallelli L, Lauria G, Cione E. Ketogenic Diet and microRNAs Linked to Antioxidant Biochemical Homeostasis. Antioxidants (Basel). 2019 Aug 2;8(8):269. doi: 10.3390/antiox8080269), perhaps to be added also in the elegant figure 1

As suggested by the Reviewer, we have now added information on the potential anti-inflammatory and antioxidant effects of ketogenic diets, also citing two studies (PMID: 25610820, PMID: 25426472) which confirm these effects in athletes – which are the focus of the present review. We have also added these effects in Figure 1 (please see the upper panel on the left side of the Figure).

It was pointed out that the decrease in muscle mass is at least in part due to the depletion of glycogen and consequent loss of water, which is however recovered by reintroducing carbohydrates. 

As nicely summarized by the Reviewer, glycogen depletion might contribute to explain the reduction in muscle mass observed with ketogenic diets. However, this would be recovered by reintroducing carbohydrates. This is the reason why in Section 4 (‘Perspectives’) the potential benefits of carbohydrate reintroduction before competitions (not only for muscle mass, but also for performance) is discussed. Nonetheless, we recognize the existence of some controversy in this issue. We have now included two more studies on this topic to further explain this issue:

  • Kysel, P.; Haluzíková, D.; Doležalová, R.P.; Laňková, I.; Lacinová, Z.; Kasperová, B.J.; Trnovská, J.; Hrádková, V.; Mráz, M.; Vilikus, Z.; et al. The influence of cyclical ketogenic reduction diet vs. Nutritionally balanced reduction diet on body composition, strength, and endurance performance in healthy young males: A randomized controlled trial. Nutrients 2020, 12, 1–12.
  • Michalczyk, M.M.; Chycki, J.; Zajac, A.; Maszczyk, A.; Zydek, G.; Langfort, J. Anaerobic performance after a low-carbohydrate diet (LCD) followed by 7 days of carbohydrate loading in male basketball players. Nutrients 2019, 11, 1–13.

I would also underline that if in a low-calorie regime the anorectic effect is an advantage in terms of research of hypertrophy, it is a disadvantage, so even in ad libitum regimes you are not able to take enough calories

We completely agree with the Reviewer’s comment. We have now discussed in further detail that, due to the appetite suppressant effects of ketosis, ad libitum ketogenic diets might result in a reduced energy intake, which in turn could have detrimental effects on muscle mass.

As for the mTOR pathway, a leucine supplement could be useful, as it is a direct and insulin-dependent activator of the pathway.

We have now stated in Section 4 (‘Perspectives’) that protein/amino acid supplementation (e.g., by means of a leucine supplement) could prevent the detrimental effects of ketogenic diets on muscle mass.

Finally, while probably not the ideal scheme to promote hypertrophy, it could be important to establish an effective protocol (perhaps alternating carbohydrate refills) for long-term periods of ketogenic dictated by pathological states such as epilepsy, migraine, or lipedema.

Thank you for highlighting this important point. The potential role of including carbohydrate refills during ketogenic diets it is now discussed in Section 4 (‘Perspectives’) as follows:

Finally, a major concern with KD is their long-term safety (Joshi, Ostfeld, & McMacken, 2019). KD have been overall reported to be safe and to improve different risk factors of cardiovascular diseases such as obesity and glucose metabolism, although the long-term sustainability of KD-induced benefits remains unclear (Kosinski & Jornayvaz, 2017; Ludwig, 2020). Moreover, a great proportion of individuals starting KD reports several symptoms (known as ‘keto flu’) during the first weeks including headache, fatigue, nausea, dizziness and gastrointestinal discomfort (Bostock, Kirkby, Taylor, & Hawrelak, 2020). There is also evidence of increased levels of low-density lipoprotein cholesterol and apo-B–containing lipoprotein with this type of diet (Retterstøl, Svendsen, Narverud, & Holven, 2018). It should be taken in mind that, as with any diet (including low-fat diets), the quality of the nutrients ingested (e.g., processed vs unprocessed foods, refined vs unrefined carbohydrates, saturated vs unsaturated fats) should be a primary focus (Ludwig, 2020). In this regard, given that KD are characterized by a high intake of saturated fats or animal-based foods as well as a low fiber intake, which could be detrimental for cardiovascular health, their substitution by polyunsaturated fats and plant-based foods (e.g., including fats from avocado, nuts, coconut or olive oil, and proteins from tofu, pea, tempeh or seitan) might be recommendable (Hooper et al., 2020; Kim et al., 2019).

However, although the aforementioned strategy has yielded promising results in some studies, evidence is still scarce and another study applying a cyclical ketogenic diet (i.e., 5 days of ketogenic diet and 2 of carbohydrate reintroduction) [Kysel et al., 2020, PMID: 32947920] found detrimental effects on performance and muscle mass compared with a conventional western diet. Thus, as stated in our manuscript, we believe that the currently available evidence does not allow designing effective evidence-based protocols for athletes, which are the focus of the present review.

REFERENCES

Bostock, E. C. S., Kirkby, K. C., Taylor, B. V., & Hawrelak, J. A. (2020). Consumer Reports of “Keto Flu” Associated With the Ketogenic Diet. Frontiers in Nutrition, 7(March), 1–6. https://doi.org/10.3389/fnut.2020.00020

Hooper, L., Martin, N., Jimoh, O. F., Kirk, C., Foster, E., & Abdelhamid, A. S. (2020). Reduction in saturated fat intake for cardiovascular disease. Cochrane Database of Systematic Reviews, 2020(8). https://doi.org/10.1002/14651858.CD011737.pub3

Joshi, S., Ostfeld, R., & McMacken, M. (2019). The Ketogenic Diet for Obesity and Diabetes— Enthusiasm Outpaces Evidence. JAMA Internal Medicine, 179(9), 1163–1164. https://doi.org/10.1136/bmj.i2716

Kim, H., Caulfield, L. E., Garcia-Larsen, V., Steffen, L. M., Coresh, J., & Rebholz, C. M. (2019). Plant-Based Diets Are Associated With a Lower Risk of Incident Cardiovascular Disease, Cardiovascular Disease Mortality, and All-Cause Mortality in a General Population of Middle-Aged Adults. Journal of the American Heart Association, 8(16). https://doi.org/10.1161/JAHA.119.012865

Kysel, P.; Haluzíková, D.; Doležalová, R.P.; Laňková, I.; Lacinová, Z.; Kasperová, B.J.; Trnovská, J.; Hrádková, V.; Mráz, M.; Vilikus, Z.; et al. The influence of cyclical ketogenic reduction diet vs. Nutritionally balanced reduction diet on body composition, strength, and endurance performance in healthy young males: A randomized controlled trial. Nutrients 2020, 12, 1–12.

Kosinski, C., & Jornayvaz, F. R. (2017). Effects of ketogenic diets on cardiovascular risk factors: Evidence from animal and human studies. Nutrients, 9(5), 1–16. https://doi.org/10.3390/nu9050517

Ludwig, D. S. (2020). The Ketogenic Diet: Evidence for Optimism but High-Quality Research Needed. Journal of Nutrition, 150(6), 1354–1359. https://doi.org/10.1093/jn/nxz308

Retterstøl, K., Svendsen, M., Narverud, I., & Holven, K. B. (2018). Effect of low carbohydrate high fat diet on LDL cholesterol and gene expression in normal-weight, young adults: A randomized controlled study. Atherosclerosis, 279, 52–61. https://doi.org/10.1016/j.atherosclerosis.2018.10.013

On behalf of all authors, many thanks for this insightful review.

Round 2

Reviewer 1 Report

The authors have made several improvements to the paper.
Unfortunately, there are still many critical issues that prevent the paper from being published.
Dietary prescription, in order to have any hope of achieving results, must have a personalized approach with long-term effects. 
A ketogenic diet, due to its strongly hypocaloric setting, is necessarily of short duration and can be useful in subjects who need to lose weight quickly (e.g. bariatric patients). 
As the authors themselves conclude there are no studies on the efficacy of this diet in the long term. In subjects who practice anaerobic sports with the objective of increasing muscle mass, a strong caloric reduction has a depressing effect on the amount of muscle mass but a high effectiveness on the loss of fat mass. It is therefore evident that the efficacy of this type of diet cannot be limited and that therefore the design of the study is incorrect.